# Influence of Composition and Plasma Power on Properties of Film from Biodegradable Polymer Blends

**DOI:** 10.3390/polym12071592

**Published:** 2020-07-17

**Authors:** Leona Omaníková, Ján Bočkaj, Mirko Černák, Roderik Plavec, Jozef Feranc, Patrik Jurkovič

**Affiliations:** 1Institute of Natural and Synthetic Polymers, Faculty of Chemical and Food Technology, Slovak University of Technology, Radlinského 9, 812 37 Bratislava, Slovakia; jan.bockaj@stuba.sk (J.B.); roderik.plavec@stuba.sk (R.P.); jozef.feranc@stuba.sk (J.F.); patrik.jurkovic@stuba.sk (P.J.); 2CEPLANT, Department of Physical Electronics, Faculty of Science, Masaryk University, Kotlarska 2, 611 37 Brno, Czech Republic; cernak@physics.muni.cz

**Keywords:** plasma, biodegradable polymer blends, polylactic acid, polyhydroxybutyrate, design of experiment

## Abstract

The work is focused on the study of surface plasma treatment (DCSBD) of films from biodegradable polymers from renewable sources based on polylactic acid (PLA) and polyhydroxybutyrate (PHB). A 4-factor design of experiment was used where the selected variable parameters were the plasma device power, the time of plasma treatment, the ratio of PHB in the polymer blend with PLA, and the content of acetyl tributyl citrate (ATBC) plasticizer in the PLA + PHB blend. The surface total energy and the polar component were evaluated immediately after surface plasma treatment and after 5 h of sitting. Topography of foil surfaces was also studied by AFM. In terms of plasma power and activation time, the greatest increase in surface energy values was observed with a short plasma time of 2 s and a high power of 400 W. Increasing the content of ATBC in interaction with the high concentration of PHB in the blend results in a reduction in the difference of both the polar component and the total free surface energy.

## 1. Introduction

Recently, there have been efforts to develop and manufacture new polymeric materials, which would, by their application and processing properties, satisfy the requirements of both the manufacturer and the consumer, while they would not impose a high burden on the environment. The combination of biodegradability and materials from renewable resources is a unique possibility to reconcile the whole life cycle of plastic material with the natural cycle, which means that plastics made from natural sources return to nature where they are fully assimilated without negatively impacting the environment. Bioplastics from renewable resources represent a new generation of plastics that further reduces the environmental effect of plastic waste. The replacement of petroleum polymeric materials with biodegradable plastics will allow the conservation of fossil resources and the reduction of carbon monoxide in emissions, which will significantly contribute to improve the quality of the environment. Biodegradable polymers are increasingly used in both the medical and the packaging industries. The main reason for this is the presence of flexible ester bonds in the aliphatic chain, which allow degradation of the material in a different pH environment to nontoxic products. In the packaging industry, the properties of these materials such as biodegradability, renewability, transparency, excellent film-forming properties, permeation selectivity, or good thermal, mechanical, or processing properties are applied. The most commonly used biodegradable polymeric materials include polylactic acid (PLA) and polyhydroxybutyrate (PHB) [1,2,3,4,5,6].

Polylactic acid is used in many sectors due to biodegradable nature. For example, in medicine as a material suitable for implants, sewing supplies, screws, or valves. In the packaging industry, PLA is used for the production of films for food, cups, bags, tea bags, and, last but not least, for composting films and bags required for agriculture [7,8].

Structurally related to PLA are the polyhydroxyalkanoates (PHA) class of polyesters derived from hydroxyalkanoic acids, which can differ in chain length and in positions of their hydroxyl groups. Just like PLA, PHAs are also derived from renewable resources such as sugars, starch, or fatty acids, however, no chemical transformation is needed. The most common representative among this class of biodegradable polymers is poly-3-hydroxybutyrate (PHB) [9]. Poly-3-hydroxybutyrate (PHB) is a 3-hydroxybutyrate homopolymer. It is a linear polyester of D (-)-3-hydroxybutyric acid. It exhibits interesting thermoplastic properties and biodegradability in compost and other environments, such as seawater, and therefore, attracts considerable commercial interest [10,11,12]. The biggest disadvantages of polyhydroxyalkanoates PHAs are the relatively high processing temperatures and brittleness. In order to avoid these effects during manufacturing, other polymers and plasticizers are added to PHAs. The most commonly used plasticizers include glycerol, acetyl tributyl citrate, salicylic and acetylsalicylic esters, polyethylene glycol, and others [13,14].

As a rule, individual biopolymers do not have the full spectrum of parameters necessary for their optimal processing and subsequent application. In order to achieve good processability, thermal stability, and suitable physical–mechanical or dynamic–mechanical properties of the resulting materials, different types of additives are added to the biopolymers and often the individual polymers are mixed with each other. This results in polymer blends which combine the properties of the individual components while increasing the ultimate variability in the properties of the polymeric materials. Acetyl tributyl citrate is used as the additive for polyhydroxybutyrate. It is a derivative of natural citric acid and is nontoxic. It has a significant effect on the thermal properties of PHB, but in order to achieve the required properties and the degree of crystallization, the samples need to be cooled very quickly below the melting point. When mixed with PHB, it generally functions as a nucleating agent. By regulating the size and the amount of the crystallites by using a nucleating agent, it is possible to achieve greater flexibility and transparency of the produced material [15,16].

The preparation of PLA/PHB blends is a simple way to simultaneously improve PLA and PHB properties, e.g., the addition of PHB increases crystallinity and improves barrier properties and hydrophobicity of PLA, whereas PLA in PHB reduces its fragility and improves processability. However, PLA/PHB blends still remain fragile, so it is necessary to add a plasticizer to the system that improves their processability, shape stability, and the flexibility needed to produce the films. One type of plasticizer used in the manufacture of a PLA/PHB blends is acetyl tributyl citrate (ATBC). This natural plasticizer is water insoluble and can be used in the manufacture of films and packaging for food and toys [17]. The main disadvantage of these materials is the hydrophobic surface that is characterized by low surface energy, which makes them unsuitable for some applications such as bonding and printing of polymer films. For this reason, their surface modification is necessary. In addition to the introduction of new environmentally friendly polymeric materials, it is also necessary to use technologies for their processing or subsequent surface treatment. 

For polymer surface modifications, it is possible to apply different methods such as plasma treatment techniques. With these techniques, it is possible to selectively modify chemical and physical properties without affecting the bulk characteristics of the materials. Plasma is a highly ionized gas consisting of ions, radicals, excited molecules, and free electrons, and it preserves its electrical neutrality. Low temperature plasma can be created by electrical discharges in low-pressure gases, and it can alter several surface properties of polymers, such as adhesion, penetrability, wettability, and biocompatibility, usually without modifying their bulk properties. Effects could be either individual or combined based on the applied plasma conditions. In a typical modification process, the initial step involves breaking the existing chemical bonds and forming new ones, resulting in new groups at the surface, which then undergo further oxidation as the surface is exposed to the air. These changes are due to different processes, including oxidation, degradation, and cross-linking, and their effectiveness depends on plasma conditions and polymer surface chemistry [18]. Surface of polymers, due to the oxidation processes, may bond with other chemical agents [19]. It is well known that the treatment of plasma can disrupt chemical bonds on the surface of the material and incorporate new oxygen-containing polar groups such as, e.g., -COOH, -O-OH, -OH, and so on. Thereby increasing hydrophilicity and surface energy of the polymeric material as well as roughness [20]. The topography of polymeric surface can also be modified by means of plasma irradiation due to its partial etching or degradation process. As a result of modified chemical composition, the wetting characteristics and surface adhesion of polymeric materials can also be changed. The attractiveness of plasma treatment amongst other conventional surface modification techniques lies in the ability to alter only surface properties, up to several nanometres deep, without affecting the bulk characteristics of materials. Moreover, short-time plasma irradiation does not overheat materials, so their destruction may be avoided [21]. Hydrophobicity (water repelling) and hydrophilicity (water attracting) is an important function of surface phenomena. Unlike metal, polymer surfaces are mostly hydrophobic because of low polarity, dispersive forces, and low surface energy. Although hydrophobic polymers have the vast majority of applications, modification to hydrophilic nature is very important for liquid attraction, extraction, wetting, and spreading. Hydrophilic properties are equipping surface interface, which is not possible in case of hydrophobic polymers. Some common polymers such as poly(tetrafluoroethylene) (PTFE), polyvinylidene fluoride (PVDF), and polydimethylsiloxane (PDMS) exhibit superior hydrophobicity, and their applications are also limited. Modification of such surfaces reduces hydrophobic nature and improves surface characteristics, which fit for extended applications [22].

The diffuse coplanar surface barrier discharge (DCSBD) due to its low thermal impact, i.e., minimal damage caused to the treated material and mentioned hardware flexibility, has found a wide range of applications in the treatment of polymers, paper, textiles, ceramics, glass, metal, silicon, electronic devices, and common low-added-value materials. Among the mentioned advantages of DCSBD, there are some of high importance: capability of working at atmospheric pressure in air and various gases, high speed of the treatment, possibility of the roll-to-roll arrangement, low cost of the equipment, and suitability for in-line industrial processing [23]. DCSBD plasma has unique properties, which clearly differentiate it from other atmospheric pressure plasma sources. DCSBD enables to generate a thin uniform plasma layer with high power density in any working gas. The main advantages of the DCSBD for surface treatment are the relatively low applied voltage and the high energy density and efficiency. The effect of plasma on surfaces is brought on by the concurrent reactions of numerous excited and ionized gas phase species, radicals, and UV radiation [24]. Plasma modification of polymer surfaces has been shown to be an equivalent alternative to wet chemical surface treatment routes for several reasons: universality, reproducibility, short reaction time, and environmental safety.

Plasma devices employing low temperature plasma are suitable for surface treatments of biodegradable polymeric materials, because they do not cause any thermal damage to the surface of the material and thus, can be used for chemical and physical treatments of polymers. One of the first studies on plasma modification of biodegradable polymers was published in 1997 by Hirotsu et al. These authors treated PLA fabrics with a low-pressure RF glow discharge sustained in pure oxygen and nitrogen and found that when a PLA fabric was treated with a glow discharge plasma, the fabric lost weight and was thus etched by the plasma. Moreover, weight loss measurements clearly showed that oxygen plasmas had a more pronounced etching effect than nitrogen plasmas and that the weight loss degree increased with increasing discharge power. It was also found that 10 s activation of DCSBD plasma caused permanent PLA surface modification. Up to now, this method of modification has been widely explored in relation to commercially produced synthetic polymers such as polyethylene, polypropylene, and polyethylene terephthalate. Biodegradable polymer materials, such as PLA and PHB, suitable for the packaging industry, in which plasma treatment has not been investigated so much, are coming to the forefront [25]. Previously, several authors studied separately the surface modification of PLA and PHB by plasma, however, the results of modification of the polymer blend containing PLA and PHB are less known. In such case, it is necessary to take into account the different behaviour of individual types of polymers, which were activated by plasma. It has been observed that treatment with cold plasma using atmospheric air as gas significantly modified the surface of polycaprolactone (PCL) and poly(lactic acid) (PLA) films. The roughness of the treated films increased throughout plasma treatment time, while the water contact angle decreased, i.e., plasma treatment increases the wettability and hydrophilic character of the films. These results justify the obtained improvement in adhesion of starch on PCL or PLA films [26].

The objective of this work is to comprehensively explore the possibilities of improving the hydrophilic properties of films prepared from blends with different ratio of PLA, PHB, and acetyl tributyl citrate plasticizer by DCSBD within a 4-factor design of experiment. Plasma device performance, plasma activation time, PLA/PHB ratio, and plasticizer content (ATBC/PLA + PHB) were selected as variables. The change in surface properties of the films was evaluated by measuring the surface energy immediately after the plasma activation.

## 2. Materials and Methods

### 2.1. Materials

Polylactic acid Ingeo™ 4042D (PLA) from NatureWorks LLC, Minnetonka, MN, USA (with content of L isomer ≥99%) and polyhydroxybutyrate (PHB) from Biomer Germany were used as biodegradable biobased polymers. Acetyl tributyl citrate (ATBC) was used as plasticizer. Distilled water and ethylene glycol were used as reference liquid.

### 2.2. Blends Preparation

The blends of PLA/PHB/plasticizer were prepared using twin-screw extruder with screw diameter of 16 mm, L/D = 40 in configuration with three kneading zones. Extruded blends were cooled down in the water bath and granulated into small pellets.

The temperature profile of blending:

Hopper 160 °C→170 °C→180 °C→190 °C→190 °C→190 °C→185 °C→180 °C→170 °C die

### 2.3. Films Preparation

For films preparation, single-screw Brabender extruder and Chill Roll technology (screw diameter 19 mm, L/D = 25, and melt temperature 170 °C) was used.

### 2.4. DCSBD (Diffuse Coplanar Surface Barrier Discharge)

DCSBD generate a thin (on the order of 0.1 mm) layer of plasma with a high power density in the immediate vicinity of the treated surface and bring it into a close contact with the treated surface. A schematic diagram of a DCSBD electrode system designed for ambient air plasma generation is shown in Figure 1. Two systems of parallel strip-like electrodes (1.8 mm wide, ~0.1 mm thick, 230 mm long, and 0.4 mm strip to strip; silver) were embedded in 96% alumina using a green tape technique. The thickness of the ceramic layer between the plasma and electrodes was 0.4 mm. A sinusoidal high frequency high-voltage ~10–20 kHz, up to 15 kV peak to peak, was applied between the electrodes [25]. The produced films were subjected to plasma surface treatment at time intervals from 2 to 10 s with a unit step of 2 s at a plasma power from 300 to 400 W with a unit step of 25 W.

### 2.5. Atomic Force Microscopy (AFM)

Additionally, changes in the surface morphology were observed by scanning atomic force microscopy (AFM). Atomic force microscopy is a microscopic technique used for three-dimensional imaging of surfaces. The AFM technique can be used not only for imaging but also for creating structures or processing surfaces in the nanometre range. The movement of the probe as it passes over the sample is detected. In the experimental part of the work, an AFM microscope of the Veeco DI CP-II brand with a 5 μm scanner operating in semicontact (tapping) mode was used. An IP AutoProbeImage 2.1.15 as the evaluation program for roughness determination and Di SPM Lab as a program for editing 2D and 3D images from AFM analysis were used.

### 2.6. Surface Energy

Due to the difficulty of directly determining the free surface energy of a solid phase, indirect methods are often used. Indirect methods include measuring the contact angle of wetting. The measurement of the contact angle can be easily performed by determining the angle of the tangent of the liquid droplet with the solid surface. The contact angle of a drop on a solid surface is defined by the decrease in equilibrium by the action of three interfacial stresses s/g, s/l, and l/g. The determination of the free surface energy of solids from contact angles is based on Young’s relation of 1805, also known as Young’s equation [27].

Young’s equation describes the balance of forces between surface tensions at the interface of three phases:*γ_sv_-γ_sl =_ γ_lv_.cos Ф*(1)
where *γ_sv_* is interfacial free energy at the solid–vapour interface, *γ_sl_* at the solid–liquid interface, and *γ_lv_* at the liquid–vapour interface. If the contact angle is 0°, then the liquid will spread completely over the surface of the solid phase. If the contact angle is 180°, the liquid does not wet the solid surface.

A SEE system (surface energy evaluation system) with a built-in colour CCD camera for drop recording was used to measure the contact angle. When measuring the contact angle, 8 drops of water and ethylene glycol were applied to the samples in each series, each in a volume of 5 μL. The drops were then recorded with a built-in CCD camera and the contact angle was evaluated with Software 5.1. Surface energy was evaluated using the Owens–Wendt regression method.

### 2.7. Design of Experiment

When studying the change in surface properties of the prepared films, the schedule of the 4-factor design of experiment was followed, observing the influence of 4 factors, which were the weight ratio of ATBC to PLA + PHB, PHB to PLA weight ratio, plasma device performance, and ultimately, plasma activation time. All 4 factors are shown in Table 1 and values of regression coefficients in Table 2. In the design of experiment, the measured values for individual properties were evaluated by the group analysis of variance and full regression analysis. Value differences were always plotted in the result areas of the individual dependencies order to exclude the effect of measurement errors.

#### Changing Factors

**x1—varying ratio of ATBC to PLA + PHB:** ATBC content was increased from 0.111 to 0.25 with a unit step of 0.03475, median value 0.1805

**x2—varying weight ratio of PHB to PLA** from 0.093 to 0.25 was increased with a unit step of 0.039, median 0.171

**x3—varying plasma power:** power was increased from 300 to 400 W with 25 W unit step, central value 350 W

**x4—plasma surface treatment time:** time was increased from 2 to 10 s with a unit step of 2 s, central value of 6 s.

For regression analysis, the next equation was used:(2)$$\hat{y}=b_{0}+\overset{k}{\underset{i=1}{\sum }}b_{i}x_{i}+\overset{k}{\underset{i=1}{\sum }}b_{ii}x_{ii}+\overset{k}{\underset{j=i+1}{\sum }}\overset{k-1}{\underset{i=1}{\sum }}b_{ij}x_{i}x_{j}$$
where

k number of factors (4)

b_0_ represents absolute number

b_1_–b_4_ represent linear coefficients

b_11_–b_44_ represent quadratic coefficients

b_12_, b_13_, b_14_, b_23_, b_24_, and b_34_ represent interaction coefficients

s_bi_ represent mean square error of given regression coefficient

b_ki_ represent critical value of given regression coefficient

Critical values were calculated on 5% level of probability.

According to ANOVA theory, the next parameters were calculated:

F_1_ criteria Fisher-Snedecor criteria for testing of linear part of equation significance

F_2_ criteria Fisher-Snedecor criteria for testing of nonlinear part of equation significance

F_LF_ criteria Fisher-Snedecor criteria for testing of inaccuracy of regression equation

s_LF_+/- mean square error of regression estimation

s_E_+/- experimental mean square error.

## 3. Results and Discussion

### 3.1. Total Free Surface Energy

The influence of individual studied factors on the change of total free surface energy can be seen from the following figures. It was assumed that the plasma treatment of the films would increase the total free surface energy. Results, which are shown in Figure 2, Figure 3, and Figure 4, show that increasing ATBC content and plasma power leads to an increase in values the difference between the total free surface energy of the activated and nonactivated sample at factors x1 (ATBC/(PLA + PHB)) and x3 (plasma power). The most striking increase can be observed in plasma-activated samples at the highest power of the plasma device (400 W) when it contained the highest ATBC plasticizer concentration and lowest constant factor levels x2 (PHB/PLA) = −2 (0.093) and x4 (plasma activation time) = −2 (2 s). Under these conditions, an increase in surface energy of 15 mJ/m^2^ compared to the inactivated sample was observed at 2 s plasma activation (Figure 2). A similar trend was also observed for the mean values of constant factors x2 (PHB/PLA) = 0.250 and x4 (plasma exposure time) = 10 s, with long plasma exposure and low plasma device performance.

The most significant decrease in the difference in surface energy values was observed in samples with a high content of ATBC plasticizer, which was an undesirable effect. This fact could be caused by the interaction of the increasing concentration of PHB in the blend and the time of plasma treatment, because at their highest values, the most significant decrease of the difference in surface energy values was observe (Figure 4).

For a more accurate study of the plasma activation effect, the difference of the polar surface energy component of the activated and nonactivated sample of PLA/PHB blends was evaluated according to the factors under consideration. The result areas of the polar free surface energy component are shown on Figure 5, Figure 6, and Figure 7, and trends similar to the total free surface energy result areas can be observed, confirming the assumption that the contribution of the polar component has the most significant effect on the increase in total free surface energy. Additionally, in this case, the highest increase in the polar surface energy component was observed on samples containing the highest concentration of ATBC plasticizer and activated at a high power of 400 W and the exposure time of 2 s of the plasma device. The polar component increased by 25 mJ/m^2^ (Figure 5) compared to samples without plasma treatment. Increasing the plasma treatment time to 6 s (centre of experiment, constant factors x2 (PHB/PLA) = 0.171 and x4(plasma treatment time) = 6 s) resulted in a milder and more uniform increase in the polar component (Figure 6) depending on the plasticizer concentration and plasma power. Another increases in plasma treatment time to 10 s (at constant factor values x2 (PHB/PLA) = 0.25 and x4(plasma activation time) = 10 s) led to a decrease of the polar component of surface energy both with increasing concentration of ATBC and decreasing of plasma power. Increasing the concentration of ATBC in interaction with a higher concentration of PHB in the blend (factor x2 (PHB/PLA)) led to in a reduction in the difference of both the polar component and the total free surface energy. A comparison of the graphical dependencies 2 to 4 and 5 to 7 shows that the polar component contributes to the increase in surface energy, as it is was observed with the same tendencies of graphical dependencies. As was shown by the results of the experiment, plasma interacts with the individual components of the polymer blend (PLA, PHB, and ATBC), while the possible migration of the plasticizer to the surface of the polymer film is taken into account. The high concentration of ATBC as well as the PHB content of the blend does not lead to the desired result.

Total surface energy was increased the most for samples which were activated at high power of 400 W and short plasma treatment time of 2 s. Long-time plasma treatment (10 s) may also result in the interaction of the newly built polar groups with each other and the cleavage of low-molecular-weight substances from the surface of the film, thus explaining the lower surface energy values.

### 3.2. Total Free Surface Energy Evaluated after 5 h of Plasma Treatment

The change in the total free surface energy difference after 5 h of plasma treatment was also studied. In Figure 8, Figure 9, and Figure 10, total free surface energy result areas measured 5 h after activation are observed depending on factors x1 (ATBC/(PLA + PHB)) and x3 (plasma power) at a constant factors x2 (PHB/PLA) and x4 (plasma exposure time). At the lowest values of the concentration of PHB and the time of plasma discharge exposure, a decrease in the total free surface energy difference with increasing ATBC content and decreasing plasma power was observed in its entirety. At the mean values of constant factors, the local minimum is visible in the middle of the result area (Figure 9). At high concentration of PHB and plasma exposure time, an increase of total surface energy difference is observed with increasing ATBC content and decreasing plasma discharge performance (Figure 10). Compared to the results immediately after plasma treatment, a steep increase in surface energy values was observed, particularly with increasing plasma power in samples containing very low ATBC content. The difference in the surface energy values of the plasma and nonplasma films was evaluated as 18 to 35 mJ/m^2^ (Figure 8). Plasma-activated polymer films represent a dynamic system. It is believed that during the aging process, different reactions occur between the free radicals on the surface of the film and elements of the environment in which the sample is stored. Another possibility is the reaction of free radicals with built-in functional groups or cross-linking. In this case, it was assumed that in the first 5 h after the plasma treatment of the samples, free radicals could react with the atmospheric oxygen, thus explaining the increase in surface energy. Only after some time, the equilibrium associated with hydrophobicity stabilize was observed. The high concentration of ATBC plasticizer in interaction with plasma treatment adversely affects the surface energy. This may also be due to the possible migration of the plasticizer to the film surface. Due to low molecular weight of ATBC and three hydrophobic butyl groups and acetyl groups in its structure, plasticizer can migrate during storage on the surface of polymer film.

Prolonged exposure of plasma appears to cause a reaction between plasma and ATBC on the surface, and other low-molecular-weight gaseous substances can be formed, which may leave the surface of the film. Plasma activation has been found to be less effective in high PHB samples in terms of increase in surface energy. Although both PLA and PHB belong to the group of polyesters, they exhibit partially different degradation behaviour under the influence of relative humidity, temperature and UV radiation [28,29].

In general, both PLA and PHB degrade by hydrolysis. Results of the hydrolysis are cleavage of ester bonds and consequently, chain tearing. Plasma activation leads to degradation in the surface layers of the material as a result of UV radiation, when the photooxidation reaction produces hydroperoxides and causes degradation to form compounds containing carboxyl groups and terminal diketone groups [30].

PHB is less stable to degradation reactions, which may be the reason that in samples with a high PHB content, the increase in surface energy due to plasma activation was lower.

#### AFM

A change in the chemical composition of PHB/PLA surfaces as a result of plasma treatment was reflected also in a change in physical properties such as surface roughness. After plasma treatments, the surface turned rough as the sign for plasma etching or removal of the surface layer. During plasma treatment, the sample surface is exposed to various particles. Oxidation leads to a formation of oxygen-rich functional groups as well as formation of low mass volatile molecules. They interact with the surface both chemically and physically. The chemical interaction causes oxidation of the surface. Plasma technique can be easily used to induce the desired groups or chains onto the surface of a material. Therefore, plasma treatment was a convenient method to improve the hydrophilicity of the polymer. Yang et al. argued that the increase in roughness is one of the reasons for the increase in hydrophilicity in plasma-treated samples, since the surface wettability is also an effect of the surface roughness. However, roughness is not the only factor affecting the increase in hydrophilicity. The formation of polar groups such as COO− and CO on the surface also contributes to the increase in hydrophilicity [31]. Selected samples were analysed by AFM to assess the effect of plasma equipment performance and plasma treatment time on film surface character.

In Figure 11, two film samples with the same composition of PHB and PLA, which were activated at two different powers (325 W for sample 3 and 375 W for sample 8) for 4 s, are compared. The mean quadratic roughness of the nonactivated films by plasma was RRMS = 1.2 nm. After plasma treatment, it increased to RRMS = 4.2 nm for a sample activated at a lower plasma performance of 325 W and to RRMS = 7.0 nm for a sample activated at a higher performance of plasma equipment of 375 W.

Similarly, in Figure 12, AFM images of nonactivated surfaces and surfaces after plasma treatment at 375 W, but for various plasma time, i.e., time 4 s for sample 6 and 8 s for sample 13, are shown. Additionally, in this case, an increase in roughness was observed from RMS = 1.2 to 2.5 nm (Figure 12B) for the 4 s plasma time treatment and to RMS = 6.3 nm for the 8 s plasma time (Figure 12D). Increasing the plasma power as well as the plasma treatment time leads to roughening of the surface. However, these values are also influenced by the composition of the polymer blends from which the films are prepared. Although samples 6 and 8 were activated under the same conditions (375 W, 4 s), their roughness varies. For sample 8, RMS = 7.0 and for sample 6, RMS = 2.5 nm. The two samples differ in PHB content in blend. For sample 6, within the design of experiment, the ratio PHB/PLA was 0.1325, whereas for sample 8, this ratio was higher as 0.2108. The higher roughness was observed in the samples with a higher content of polyhydroxybutyrate (PHB), which were more susceptible to degradation reactions. It is more likely that during activation, polar groups and low-molecular products from the surface of films are cleaved, resulting in increased roughness and lower surface energy values than the previous results.

## 4. Conclusions

Plasma treatment efficiency for PLA and PHB film modification was evaluated using the designed 4-factor experiment. The composition of the blend of ATBC/(PLA + PHB), PHB/PLA weight ratio, plasma device performance, and plasma treatment time were studied as variable factors. Differences of total free surface energy and polar component were evaluated as output parameters. The mathematical processing of the results showed that plasma activation makes it possible to improve the hydrophilic properties of films from biodegradable polymer blends of different compositions. It was found that the total free surface energy depends primarily on the composition of the blend and the conditions of plasma treatment. The results confirmed the known knowledge from the theory that plasma treatment is not constant and the surface properties of the polymers change with time. The results were analysed immediately after plasma treatment and after 5 h of treatment. The total free surface energy after plasma activation was observed to depend on the plasticizer content and plasma power, with the most significant change that occurred at the highest ATBC content when the PHB/PLA constant was 0.093. In terms of plasma performance and plasma time, it was observed that the greatest change occurred at the lowest plasma time of 2 s and a high plasma power of 400 W. The evaluation of the polar component revealed that it contributes most to the change in total free surface energy.

Investigating the effect of plasma treatment on surface properties during aging under laboratory conditions, it was observed that total free surface energy depends not only on the plasticizer concentration but also on the PHB/PLA content, with the most pronounced change of 35 mJ/m^2^ observed in low PHB films and ATBC. Compared to the results of the surface energy evaluation immediately after plasma treatment, it was found that the surface energy difference increased over time (the difference immediately after plasma treatment was 16 mJ/m^2^ and after 5 h was 37 mJ/m^2^).

This can be explained by the fact that plasma treatment is a dynamic system and its effect changes over time depending on the activation conditions. In this case, this increase in the difference was probably due to the existence of free radicals on the surface of the film, to which atmospheric oxygen was bound after removal from the plasma and thereby incorporated more hydrophilic functional groups.

A study of the topography of plasma treatment and nonplasma treatment films showed that plasma contributes to roughening of the surface and/or increases the mean roughness values, which depend on the composition of the blend and the conditions of plasma treatment.

## Figures and Tables

**Figure 1 polymers-12-01592-f001:**
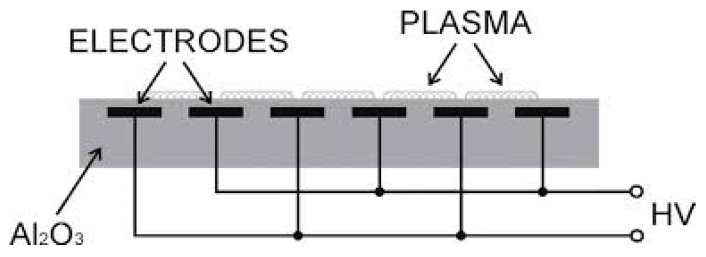
Diffuse coplanar surface barrier discharge.

**Figure 2 polymers-12-01592-f002:**
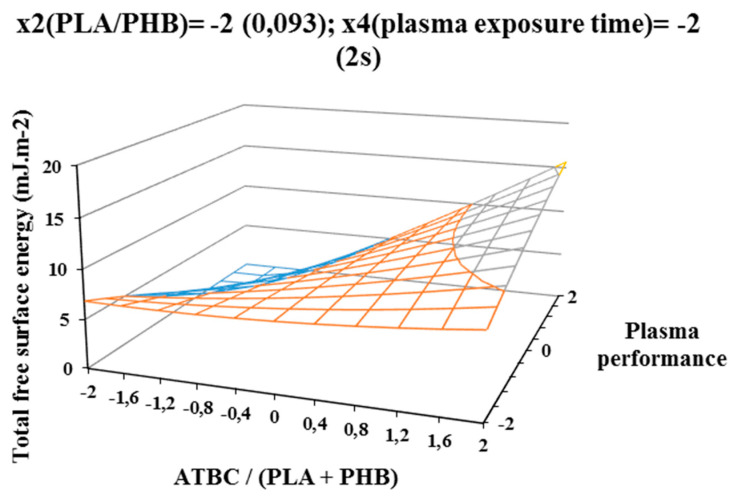
Response surface of total free surface energy difference measured immediately after plasma surface treatment in dependence of factors x1 (ATBC/(PLA + PHB)) and x3 (plasma power) at constant factor x2 (PHB/PLA) = −2 and x4 (plasma exposure time)) = −2.

**Figure 3 polymers-12-01592-f003:**
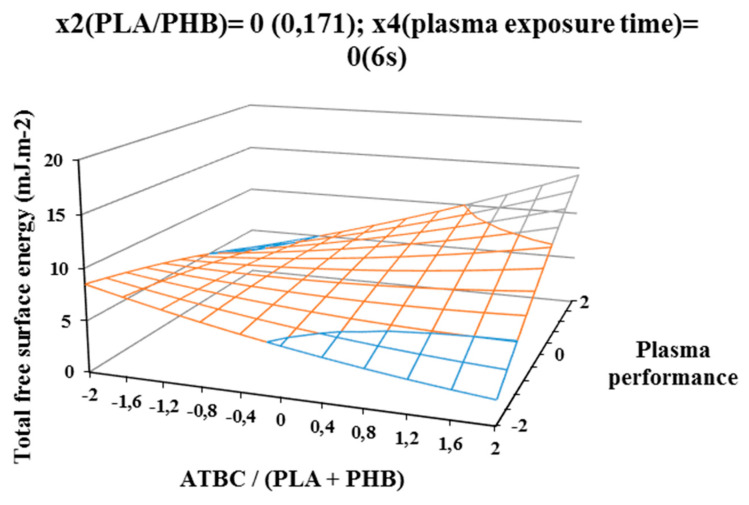
Response surface of total free surface energy difference measured immediately after plasma surface treatment in dependence of factors x1(ATBC/(PLA+PHB)) and x3 (plasma performance) at constant factor x2 (PHB/PLA) = 0 and x4 (plasma exposure time) = 0.

**Figure 4 polymers-12-01592-f004:**
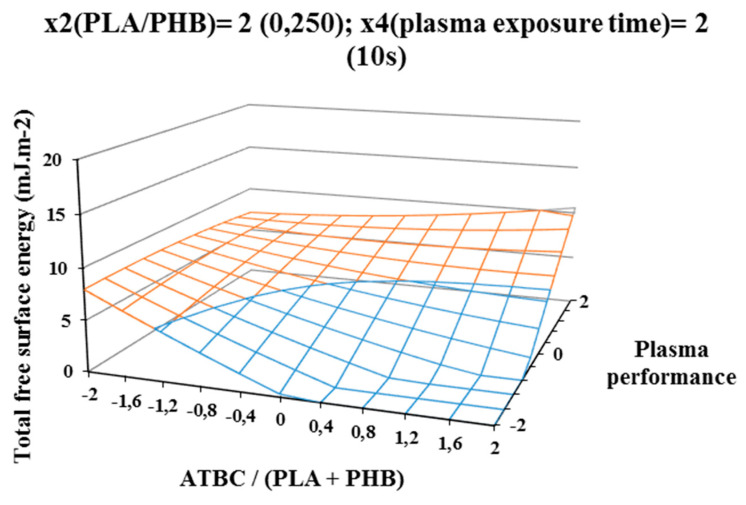
Response surface of total free surface energy difference measured immediately after plasma surface treatment in dependence of factors x1 (ATBC/(PLA + PHB)) and x3 (plasma power) at constant factor x2 (PHB/PLA) = 2 and x4 (plasma exposure time) = 2.

**Figure 5 polymers-12-01592-f005:**
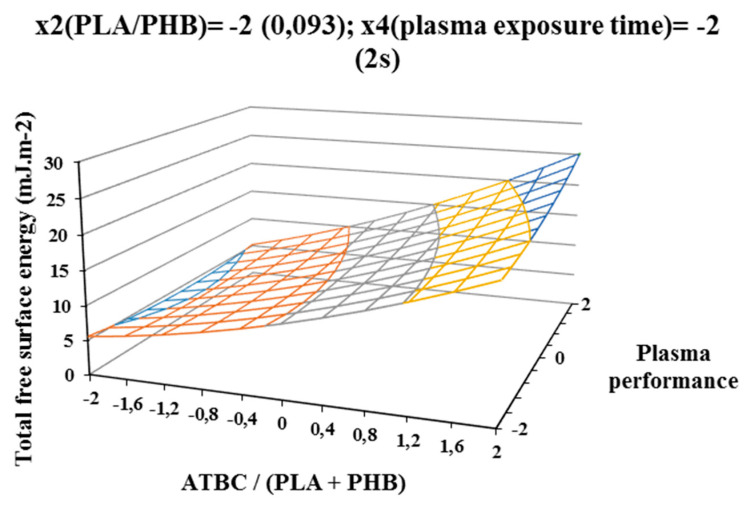
Response surface of difference of the polar component of free surface energy measured immediately after plasma surface treatment in dependence of factors x1 (ATBC/(PLA + PHB)) and x3 (plasma power) at constant factor x2 (PHB/PLA) = −2 and x4 (exposure time) plasma) = −2.

**Figure 6 polymers-12-01592-f006:**
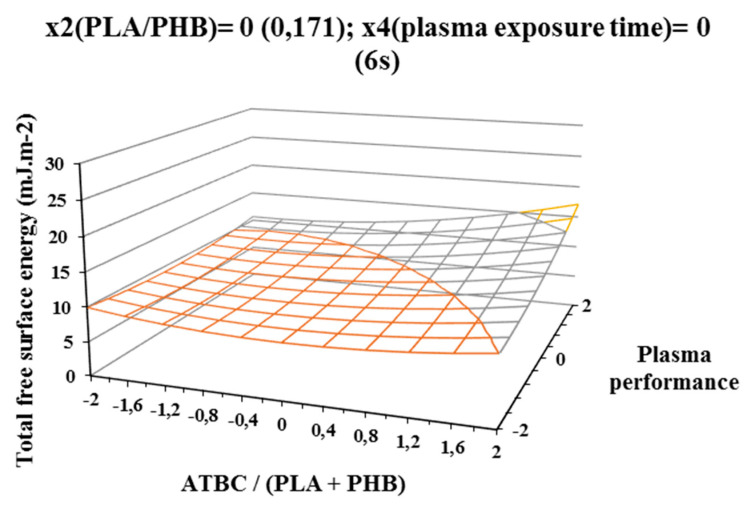
Response surface of difference of the polar component of free surface energy measured immediately after plasma surface treatment in dependence of factors x1 (ATBC/(PLA + PHB)) and x3 (plasma power) at constant factor x2 (PHB/PLA) = 0 and x4 (plasma exposure time)) = 0.

**Figure 7 polymers-12-01592-f007:**
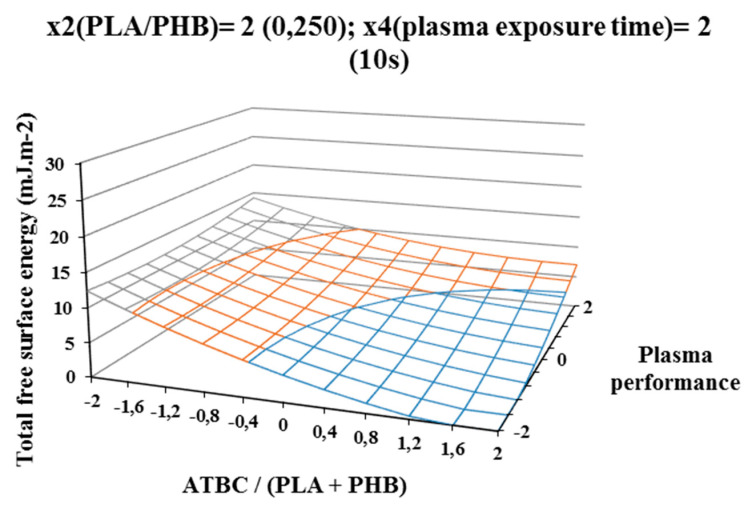
Response surface of difference of the polar component of free surface energy measured immediately after plasma surface treatment in dependence of factors x1 (ATBC/(PLA + PHB)) and x3 (plasma power) at constant factor x2 (PHB/PLA) = 2 and x4 (plasma exposure time)) = 2.

**Figure 8 polymers-12-01592-f008:**
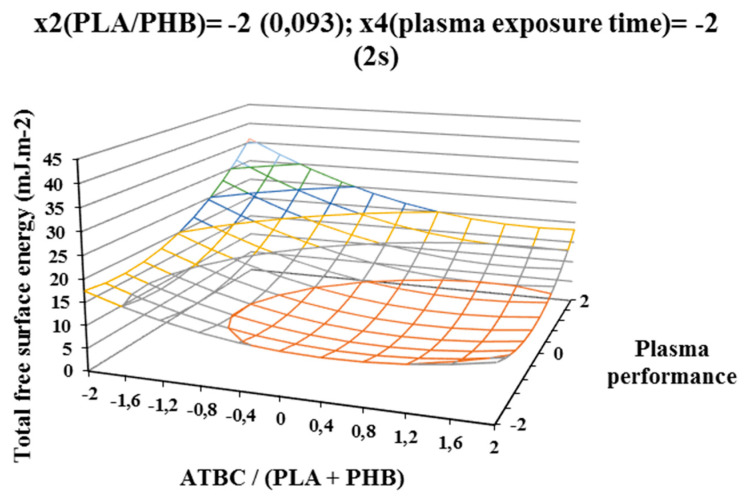
Response surface of total free surface energy difference measured 5 h after plasma surface treatment, depending on factors x1 (ATBC/(PLA + PHB)) and x3 (plasma power) at a constant factor x2 (PHB/PLA) = −2 and x4 (exposure time) plasma) = −2.

**Figure 9 polymers-12-01592-f009:**
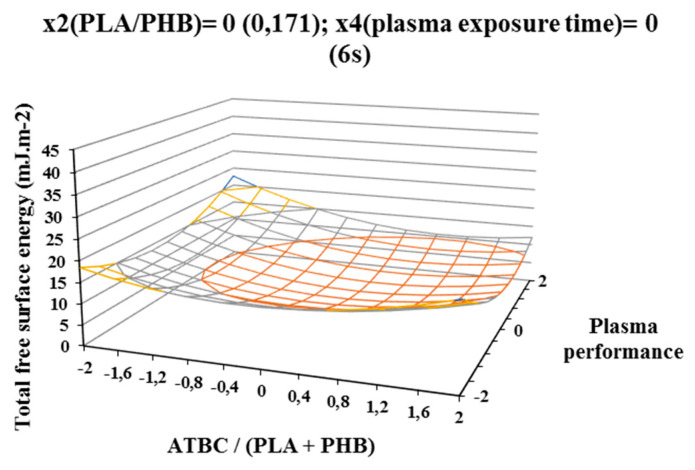
Response surface of total free surface energy difference measured 5 h after plasma surface treatment in dependence of factors x1 (ATBC/(PLA + PHB)) and x3 (plasma power) at constant factor x2 (PHB/PLA) = 0 and x4 (plasma exposure time)) = 0.

**Figure 10 polymers-12-01592-f010:**
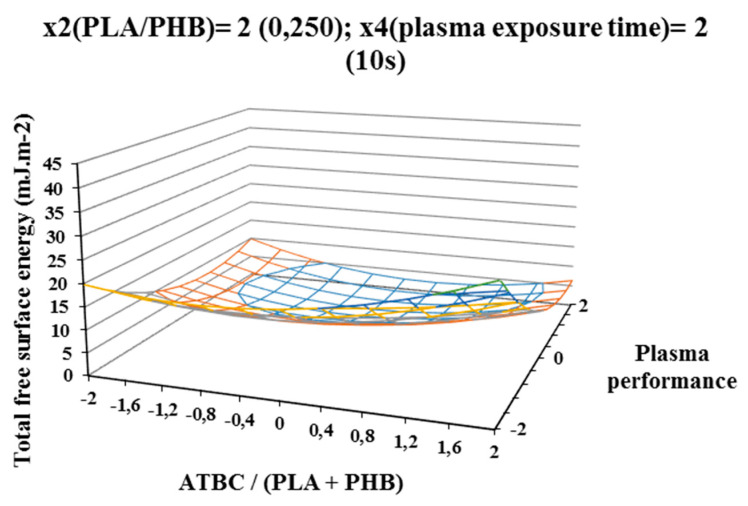
Response surface of total free surface energy difference measured 5 h after plasma surface treatment in dependence of factors x1 (ATBC/(PLA + PHB)) and x3 (plasma power) at constant factor x2 (PHB/PLA) = 2 and x4 (plasma exposure time)) = 2.

**Figure 11 polymers-12-01592-f011:**
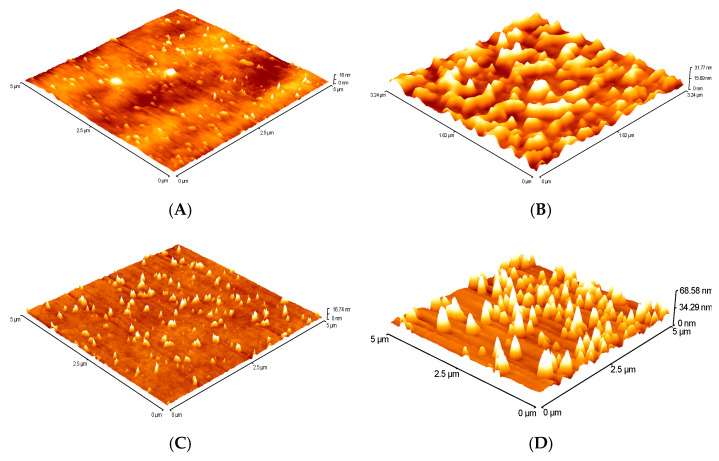
AFM images: (**A**) blend 3 nonplasma treatment, (**B**) blend 3 plasma treatment, (**C**) blend 8 nonplasma treatment, and (**D**) blend 8 plasma treatment.

**Figure 12 polymers-12-01592-f012:**
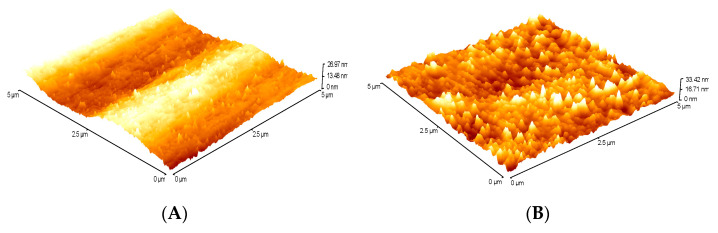
AFM images: (**A**) blend 6 nonplasma treatment, (**B**) blend 6 plasma treatment, (**C**) blend 13 nonplasma treatment, and (**D**) blend 13 plasma treatment.

**Table 1 polymers-12-01592-t001:** Factors of design of experiment.

Designation	Factor	−2	−1	0	1	2
**x1**	ATBC/(PLA+PHB)	0.111	0.145	0.180	0.215	0.250
**x2**	PHB/PLA	0.093	0.132	0.171	0.210	0.250
**x3**	Plasma power	300	325	350	375	400
**x4**	Plasma time	2	4	6	8	10

**Table 2 polymers-12-01592-t002:** Values of regression coefficients of mathematical model for (a) samples measured immediately after plasma surface treatment, (b) samples of polar component measured immediately after plasma surface treatment, and (c) samples measured after 5 h after plasma surface treatment.

**a**	**b**	**c**
	**Coefficients**	**sb**	**bk**		**Coefficients**	**sb**	**bk**		**Coefficients**	**sb**	**bk**
b0=	6.750	0.878	2148	b0=	8.913	1.219	2984	b0=	6.031	1.121	2743
b1=	0.641	0.474	1160	b1=	0.813	0.659	1611	b1=	−1.085	0.605	1482
b2=	−0.218	b2=	−0.254		b2=	−0.006
b3=	0.785	b3=	0.973		b3=	−0.526
b4=	−0.277	b4=	−1.161		b4=	−1.179
b11=	0.158	0.434	1063	b11=	0.466	0.603	1476	b11=	1.364	0.555	1357
b22=	0.249	b22=	0.298		b22=	0.794
b33=	−0.099	b33=	0.289		b33=	1.722
b44=	−0.328	b44=	−0.286		b44=	−0.086
b12=	−0.908	0.581	1421	b12=	−1.654	0.807	1974	b12=	0.356	0.742	1815
b13=	1.094	b13=	0.400		b13=	−0.790
b14=	−0.078	b14=	−0.037		b14=	0.533
b23=	0.747	b23=	0.094		b23=	−1.806
b24=	−0.199	b24=	−0.259		b24=	−0.645
b34=	−0.150	b34=	0.034		b34=	0.031
	**Value**	**Critical value**			**Value**	**Critical value**			**Value**	**Critical value**	
**F1**	1.282	4.534		**F1**	1.740	4.534		**F1**	1.939	4.534	
**F2**	0.907	4.060		**F2**	0.587	4.060		**F2**	2.421	4.060	
**FLF**	2.680	4.060		**FLF**	3.558	4.060		**FLF**	3.585	4.060	
**sLF+/-**	3.801			**sLF+/-**	6.086			**sLF+/-**	5.617		
**sE+/-**	2.322			**sE+/-**	3.226			**sE+/-**	2.966

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
