# Peer review of "Influence of Composition and Plasma Power on Properties of Film from Biodegradable Polymer Blends"

_polymers, 2020, doi:10.3390/polym12071592_

Round 1

Reviewer 1 Report

The study is too simple for polymer design. There are a lot of studies on the polymer film by plasma performance.

Reviewer 2 Report

COMMENTS [Article ID: polymers-788483]
This paper reports the study of surface plasma treatment (DCSBD) of films from biodegradable polymers from renewable sources based on polylactic acid (PLA) and polyhydroxybutyrate (PHB). The paper includes interesting results with suitable experimental design, data analysis and discussion. Therefore, it is recommended for publication in Polymers after major revision indicated below.

GENERAL COMMENTS
- Review the English grammar. There are several writing errors as: “while is take into account” (line 205).

SPECIFIC COMMENTS
Introduction
- A review of other related studies should be performed since it is important to remark the novelty of this work.

Materials and Methods
- Unify the numbers of decimals.
- Include the caption for Table 1.
- Which is the temperature for the blends preparation?

Results and Discussion
- The caption for Figure 1 is in capital letters. Rewrite it.
- Be careful with numbers since “.” should be used for decimals.
- Line 184: double space between “the” and “difference”.
- Caption of Figure 1 appears in Capital Letters.
- Explain better the correlation between the AFM results with the total free surface energy performance exhibited by the systems.

References
- Unify the format of the references following the reference style of the journal.
- Please include more updated references.
- Reference’s number 25 is in red.

Reviewer 3 Report

The paper entitled “Influence of composition and plasma performance on properties of film from biodegradable polymer blends” by Omanikova L. et al. has been reviewed. The idea of plasma treatment of biopolymer mixture (PLA and PHB) seems to be very promising. However, the article should be completely revised before publication. To start with, English should be corrected by a native speaker as there are too many mistakes that make the understanding very difficult.

The title should be more precise (you didn’t change the plasma composition, did you?).

Why do the authors search for the film hydrophilic properties? As usual, the hydrophobic properties are needed, for example, for food packaging application.

In the text it would be better to replace “plasma performance” by “plasma power”.

The complete polymer name should be given first (line 89).

There is also a mish-mash between PHA (line 45) and PHB (lines 40 and 49).

The interest in DCSBD treatment should be underlined, as whatever the source is, the plasma treatment can be used for various materials (line 101).

The L/D proportion for PLA should be given (line 111).

The description of contact angle and AFM measurements should be added in Materials and Methods.

The caption to Table 1 should be given.

Some Figures can be combined. For example, Fig. 2, 3 and 4; Fig. 5, 6 and 7; Fig. 8, 9 and 10.

The authors note a possible migration of the plasticizer to the film surface (line 247). Why? The films were prepared by extrusion, so the homogeneous distribution of ATBC should be obtained.

A possible mechanism of the plasma modification should be proposed.

The authors show the modification of surface properties (roughness and hydrophilic/hydrophobic balance). And what about polymer bulk properties? Tg value, for example?

The Reference list should be given according to Journal Requirements.

Round 2

Reviewer 1 Report

OK

Author Response

Thank you for your recension.

Reviewer 2 Report

COMMENTS [Article ID: polymers-788483]

This study is recommended for publication in Polymers after a revision of the English style all over the manuscript.

Author Response

Thank you for your recension.

Reviewer 3 Report

The authors tried to answer questions and comments that I had earlier made and revised their paper accordingly. However, there are still some questions that authors should reply and many English “bugs” that are to be corrected (especially in newly included parts). The article should be revised by the English native speaker. Moreover, the article titles are missing for some References.

The description and explication of the plasma process should be revised (lines 87-157). The authors note that “treatment of plasma…can incorporate new oxygen-containing polar groups”. That occurs in case of certain plasma, namely oxygen/air. In case of CF4 plasma, for example, the increase of the surface hydrophobicity will be observed.

The authors noted the appearance of free radicals on the surface (line 441), but no treatment mechanism was proposed. In addition, what are “unoccupied radicals” (line 537)?

Yang is not the author of Ref. 32 (lines 479-484). Besides, how can you explain that the hydrophilicity increases by means of the roughness increase? More information should be given.
